# Standardized surgical technique for single-port totally extraperitoneal inguinal hernia repair using the glove method with an intraumbilical incision

Yoshiro Imai◉*, Yusuke Suzuki, Mitsuhiro Asakuma, Yoshiharu Miyamoto, Hideki Tomiyama◉, Sang-Woong Lee

Department of General and Gastroenterological Surgery, Osaka Medical and Pharmaceutical University, Takatsuki, Osaka, Japan

* yoshiro.imai@ompu.ac.jp

## Abstract

### Background

The safety of single-port totally extraperitoneal (STEP) inguinal hernia repair compared to conventional totally extraperitoneal (CTEP) has been supported by various randomized controlled trials (RCTs). However, the optimal method remains unclear because of variations in the location and length of the incision as well as different uses of the single-port device. We standardized STEP using the glove method with a straight umbilical incision that maintained the wound profile within the confines of the umbilicus for better cosmetic satisfaction and port operative pain reduction.

### Methods

The incision length was limited to 1–1.5 cm, extending no further than the umbilicus. The STEP is performed utilizing the glove method using Alexis of XXS size. To minimize forceps interference, the surgeon dissected the spermatic cord into two distinct phases. As the mesh was inserted within the pneumoperitoneum, it was deployed safely and securely. In addition, we present the surgical outcomes at our institution for an early career surgeon.

### Results

A total of 25 unilateral inguinal hernia STEP procedures were performed between April and October 2023. The median operative time was 68 minutes, and the procedure was performed safely with no complications requiring treatment.

### Conclusion

In conclusion, STEP using the glove method with a total intraumbilical incision can be safely performed by an early career surgeon.

**Data availability statement:** All relevant data are within the paper and it's Supporting information files.

**Funding:** The author(s) received no specific funding for this work.

**Competing interests:** The authors have declared that no competing interests exist.

## Introduction

Inguinal hernia repair is one of the most commonly performed surgeries by general surgeons. In the international guidelines for groin hernia management, surgeons should provide both anterior (Lichtenstein) and posterior (laparoscopic surgery) options [1]. Both approaches did not differ in perioperative complication rates, requiring reoperation, or recurrence rates. However, laparoscopic surgery has advantages such as reduced postoperative pain, quicker recovery, and potentially faster return to normal activities [2].

Single-port laparoscopic surgery is performed in various fields owing to its pain reduction and better cosmetic outcomes than conventional laparoscopic surgery. Totally extraperitoneal (TEP) and transabdominal preperitoneal (TAPP) inguinal hernia repairs have almost equivalent surgical outcomes [2–4]; TEP is more suitable than TAPP for single-port laparoscopic surgery because it does not require suture closure of the peritoneum.

In randomized controlled trials (RCTs) comparing single-port TEP (STEP) and conventional TEP (CTEP) repair [5–11], surgical outcomes were comparable. Some studies have suggested superiority in terms of postoperative pain, quicker return to normal activities, and higher levels of cosmetic satisfaction with STEP. However, conflicting findings have been reported, with some trials not supporting these advantages. The length and location of the incision can affect postoperative pain and cosmetic satisfaction. To optimize cosmetic satisfaction and minimize postoperative pain during inguinal hernia repair, the skin incision should be kept as small as necessary for forceps entry, as specimen removal is not required.

We present our institution's standardized STEP technique that employs a straight umbilical incision to maintain a wound profile within the confines of the umbilicus, resulting in excellent cosmetic outcomes. The purpose of this paper is to assess the feasibility and safety of this technique when performed by an early career surgeon, with particular emphasis on ease of learning and patient-related outcomes.

## Materials and methods

The patients eligible for STEP are adults aged 18 or older seeking inguinal hernia repair. However, individuals with a high risk for general anesthesia, recurrent cases, and those who have undergone prostatectomy are not suitable candidates for STEP.

This study investigated various patient-related factors, including age, body mass index (BMI), sex, comorbidities, antithrombotic therapy, history of lower abdominal surgery, and hernia site and type. Additionally, perioperative outcomes were assessed, encompassing operative time, incidents of peritoneal injury, conversion rates, postoperative stay, and the occurrence of complications.

The early career surgeon started with STEP inguinal hernia repair and performed 40 cases under the guidance of the same assistant. The surgeon was started on STEP without any prior experience with CTEP. From April to October 2023, he performed 29 STEP procedures with the same assistance at the Osaka Medical and Pharmaceutical University Hospital, Takatsuki City, Japan. He consistently experienced STEP using the same glove method.

The study was conducted in accordance with the Declaration of Helsinki and its latest amendments. This study was approved by the Osaka Medical and Pharmaceutical University Ethics Committee of Takatsuki City, Japan (approval no. 2023-125:29/9/2023). All datasets were anonymized before analysis, and no personal identifiers were used. Written informed consent was obtained from all the participants after they had been provided with sufficient information regarding the purpose and procedures of the study. No minors were included.

## Operative procedure

### Extraperitoneal space dissection

Under general anesthesia, the patient was placed in the supine position with the healthy arm adducted. Subsequently, the patient was placed in the Trendelenburg position, with the side opposite to the hernia site tilted downward.

The incision length was limited to 1–1.5 cm, extending no further than the umbilicus. To minimize distortion or deformity of the umbilical region, the umbilicus was completely inverted, and a longitudinal incision was made along the midline (Fig 1). If an incision is made precisely in the middle of the umbilicus, a fascial defect called the natural orifice of the navel is observed [12]. This natural orifice is very important in single-port laparoscopic surgery, as it allows us to reach the intraperitoneal space without having to separate the fascia and insert a 5-mm port; we sequentially conducted intraperitoneal inspections to accurately diagnose and confirm the presence of any contralateral inguinal hernia. The risk of developing an abdominal incisional hernia is reduced by the use of a 5-mm port, which minimizes damage to the abdominal wall. Following intraperitoneal inspection, the 5-mm port was removed, and a Foley catheter was placed as a routine procedure for deaeration in case of accidental pneumoperitoneum. Subsequently, after dissecting the subcutaneous tissue and exposing the anterior rectus sheath on the side of the hernia, a 1.5-cm incision was made in the anterior rectus sheath. This was followed by the separation of the rectus muscle from the posterior rectus sheath. Subsequently, an Alexis (Applied Medical, Rancho Santa Mergarita, CA, USA) of XXS size was inserted in front of the posterior rectus sheath (Fig 2). This minimal incision would be difficult to perform without an Alexis of XXS size. Then, a non-powdered surgical glove (5.5 in) was placed airtightly on the wound retractor, through which three 5-mm trocars were inserted via the fingertips. A single port was completed (Fig 2). A surgical glove port is highly effective for single-port surgery [13,14]. A flexible-tip laparoscopic

**a** 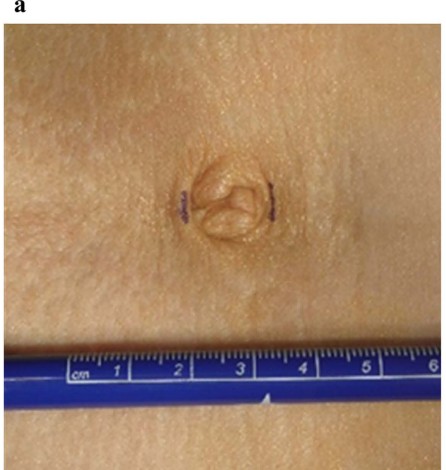     **b** 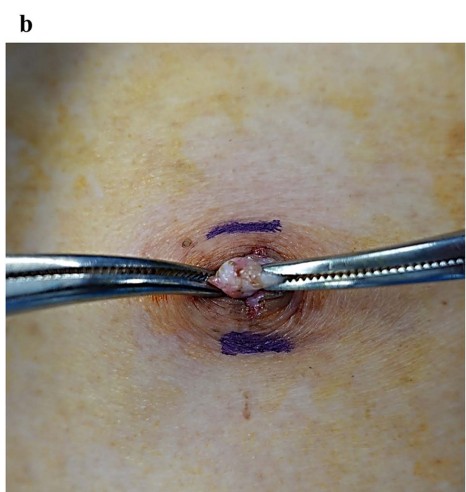

**Fig 1. Intraumbilical incision. (a)** The incision length should be limited to 1-1.5 cm, extending no further than the umbilicus. **(b)** The umbilicus is completely inverted, and a longitudinal incision is made along the midline.

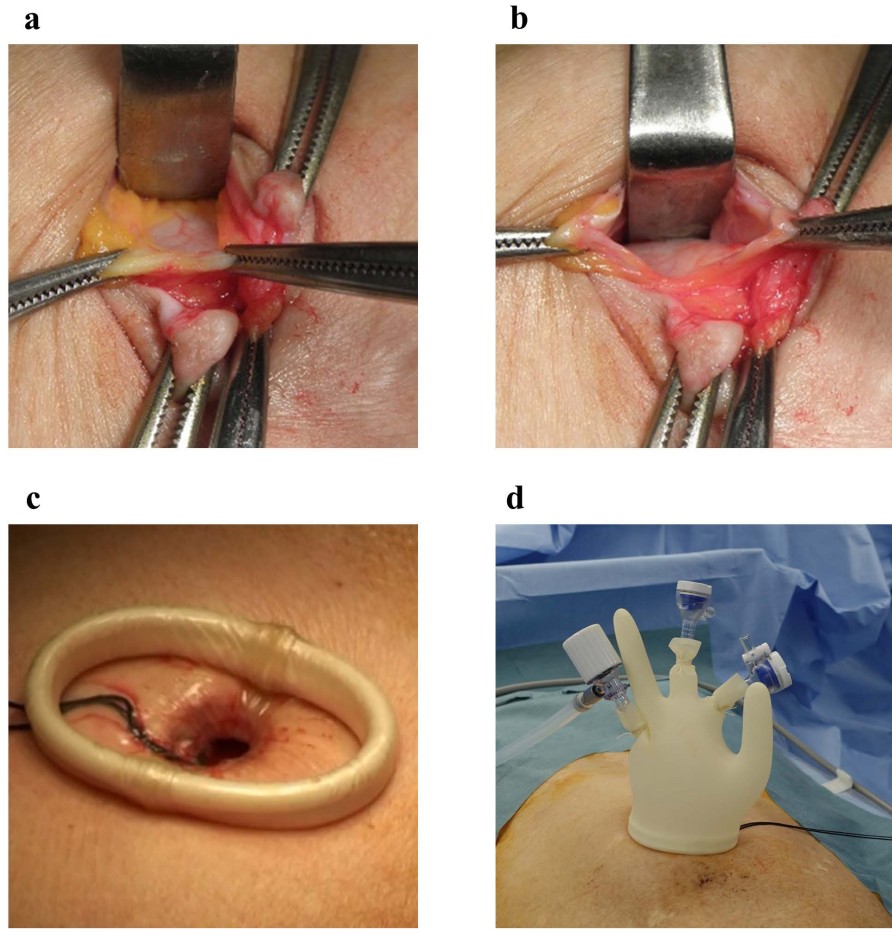

**Fig 2. Creating the surgical glove port.** (a) After dissecting the subcutaneous tissue and exposing the anterior rectus sheath on the side of the hernia. (b) This was followed by the separation of the rectus muscle from the posterior rectus sheath. (c) An Alexis of XXS size was inserted in front of the posterior rectus sheath. (d) A non-powdered surgical glove (5.5 inches) was put on the wound retractor air-tightly, through which three 5-mm trocars were inserted via the fingertips.

camera (LTFVH, Olympus, Tokyo, Japan) was inserted via the healthy side of the finger port, while the other finger ports were used for the operator's instruments. Pneumoperitoneum was induced at a pressure of 8 mmHg. Dissection of the initial extraperitoneal space was possible in direct endoscopic view without using a dissection balloon; therefore, safe and exact dissection was possible because the operation was not blinded. We invaded the Retzius cavity and exposed the Cooper's ligament. To this end, the operation was performed with only one hand, using laparoscopic diathermy scissors to minimize interference.

### Direct inguinal hernia cases

In a direct inguinal hernia, the pseudosac of the transverse fascia was confirmed using Hesselbach's triangle. Dissection is performed by grasping the hernial sac with one forceps and the pseudosac with the other, but in the cephalocaudal direction, making it relatively easy to perform with minimal interference.

## Indirect inguinal hernia cases

Indirect inguinal hernia requires parietalization of the spermatic cord. Therefore, two procedures were standardized: dissection of the spermatic ducts and the vas deferens (Fig 3).

During the dissection of the spermatic ducts, the spermatic cord was delicately grasped using forceps and slowly spread outward. A flexible-tip laparoscopic camera was inserted into the medial cavity, and the tip of the laparoscopic camera was directed toward the hernia site. The spermatic cord is visible from the medial side, and the forceps emerging from the hernial side of the screen can be observed. The horizontal intersection of the laparoscopic camera and forceps within the Alexis lumen creates a generous working space for both instruments. This space can be used for dissection using other forceps without causing interference. The hands of the surgeon and endoscopist were aligned horizontally to ensure that an adequate distance was maintained.

In the dissection of the vas deferens, forceps are used to grasp the spermatic cord and extend it in the dorsal direction. A flexible-tip laparoscopic camera was inserted in the ventral direction, with the tip of the laparoscopic camera looking down on the spermatic cord. The vertical intersection of the laparoscopic camera and forceps within the Alexis lumen creates a generous working space for both instruments. This space could be used to dissect other forceps without

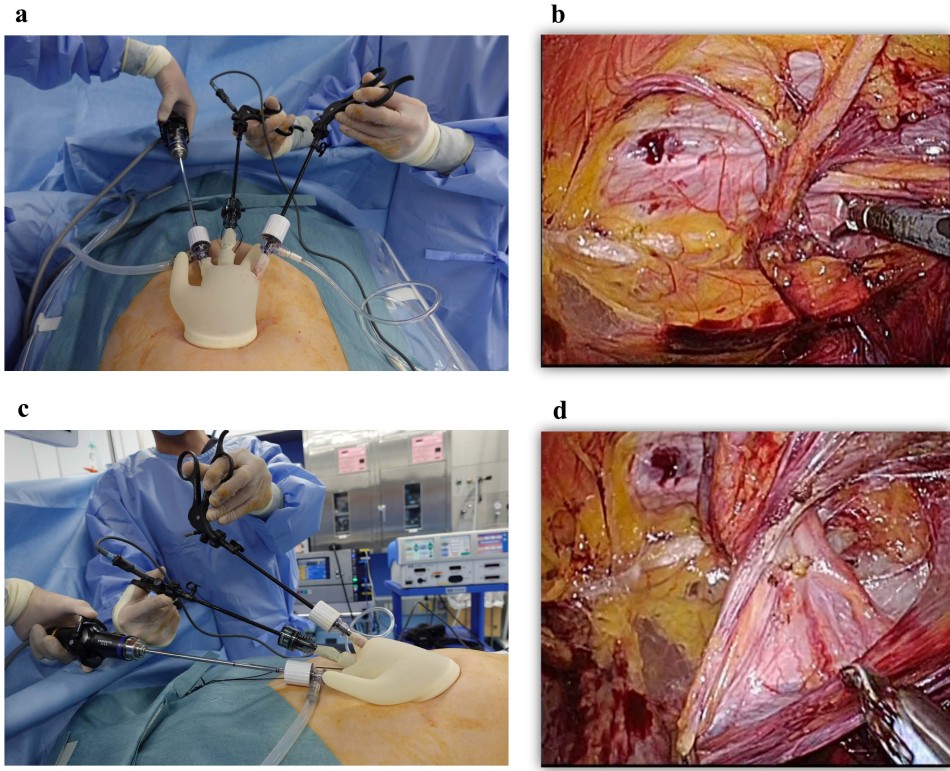

**Fig 3. Two procedures were standardized: dissection of the spermatic ducts and dissection of the vas deferens. (a)** The precise positioning of these devices creates ample working space for the laparoscopic camera and forceps within the lumen of the Alexis due to the horizontal cross configuration. **(b)** The spermatic cord is visible from the medial side, and the forceps can be observed emerging from the hernia side of the screen. **(c)** The precise positioning of these devices creates ample working space for the laparoscopic camera and forceps within the lumen of the Alexis due to the vertical cross configuration. Additionally, the forceps handles are inverted to ensure smooth vertical manipulation and ample operating space. The hands of the surgeon and the scopist are vertically aligned, ensuring that an adequate distance is maintained. **(d)** A flexible-tip laparoscopic camera is then inserted into the ventral direction, and the tip of the laparoscopic camera looks down on the spermatic cord.

interference. In addition, the forceps handles were inverted to ensure smooth vertical manipulation and ample operating space. The hands of the surgeon and the scopist were aligned vertically to ensure that an adequate distance was maintained.

**Mesh insertion and fixation**

A 3D Max Light (C.R. Bard/Davol, Warwick, RI, USA) was inserted and positioned to cover the myopectineal orifices of Fruchaud. To safely insert the mesh, forceps were used to grasp the inner portion of the mesh, and a surgical glove was covered (Fig 4). Pneumoperitoneum is created to expand the extraperitoneal cavity, thereby enabling the insertion of the mesh under direct endoscopic view with pneumoperitoneum so that it can be performed safely without being a blind operation. Because the mesh is inserted within the pneumoperitoneum without wrapping, it experiences minimal deformation and spontaneously spreads when a shape-forming mold is used. This makes it easier to deploy myopectineal orifices. The mesh was fixed to Cooper's ligament, midline, and lateral abdominal wall using SORBAFIX (C.R. Bard/Davol, Warwick, RI, USA). Furthermore, the mesh is fixed to the midline of the abdominal wall during the CTEP procedure; however, it can obstruct the suprapubic port. This problem did not occur for STEP. The lower edge of the mesh was deflated while it was secured with forceps, and the mesh was inspected intraperitoneally to ensure that it did not roll up.

## Results

The early career surgeon performed a total of 29 STEP procedures between April 2023 and October 2023. Among these, there were 25 cases of unilateral inguinal hernia and 4 cases of unilateral inguinal hernia. The demographics and clinical characteristics of patients with unilateral hernia are shown in Table 1. The average age was 75 years (range, 36–86 years), the sex was 18 males and 5 females, and the average body mass index was 22.7 kg/m$^2$ (range, 17.7 to 22.7 kg/m$^2$). Seven patients were receiving antithrombotic therapy, and five patients had a history of lower abdominal surgery.

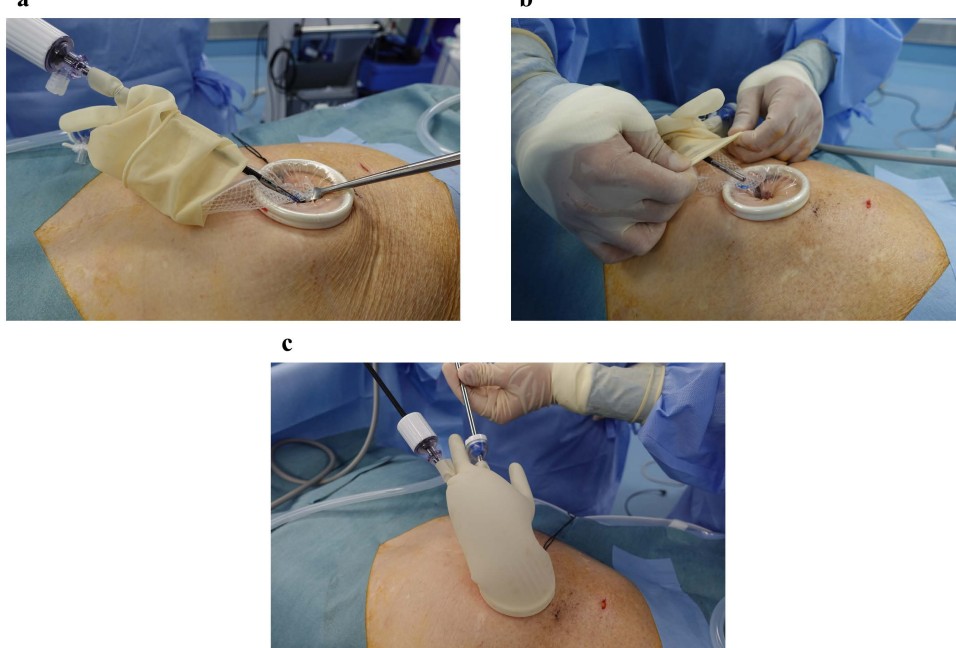

**Fig 4. Mesh Insertion.** To safely insert the mesh, forceps were used to grasp the inner portion of the mesh and a surgical glove was covered.

Six patients had a direct hernia, 15 had an indirect hernia, one had a femoral hernia, and there patients with a combined hernia. The perioperative data are shown in Table 2. The median operative time was 68 min. No conversion cases were reported, with only one instance each of seroma and hematoma. Both the conditions resolved mildly with conservative treatment. The mean duration of the postoperative stay was 1.9 days. Fig 5 shows the learning curve of STEP as performed by the early career surgeon. The operative time decreased gradually after 20 cases and stabilized after 40 cases.

## Discussion

In single-port laparoscopic surgery, the incision length and site are crucial. Several factors can influence postoperative pain and cosmetic outcomes, and the incision length and site are widely recognized as the most significant determinants.

**Table 1. Patients' characteristics.**

|  | n = 25 |
|---|---|
| Age (years) | 73 (36-86) |
| BMI (kg/m$^2$) | 22.75 (17.77-22.75) |
| Sex (Male/ Female) | 18/ 7 |
| Comorbidities |  |
| High blood pressure | 9 (36%) |
| Cardiovascular disease | 7 (28%) |
| Diabetes mellitus type 2 | 5 (20%) |
| Collagen disease | 3 (12%) |
| Antithrombotic therapy | 7 (28%) |
| Previous lower abdominal surgery | 5 (20%) |
| Site of hernia (Right/ Left) | 15/ 10 |
| Type of hernia |  |
| Direct | 6 (24%) |
| Indirect | 15 (60%) |
| Femoral | 1 (4%) |
| Direct + Indirect | 2 (8%) |
| Direct + Femoral | 1 (4%) |

BMI, body mass index.

**Table 2. Preoperative data.**

|  | n = 25 |
|---|---|
| Operative time (min) | 68 (52-89) |
| Peritoneal injury | 1 (4%) |
| Conversion | 0 |
| Postoperative stay (days) | 1.9 (1-2) |
| Complication |  |
| Seroma | 1 (4%) |
| Hematoma | 1 (4%) |
| Wound infection | 0 |
| Mesh infection | 0 |

Data are presented as n (%).

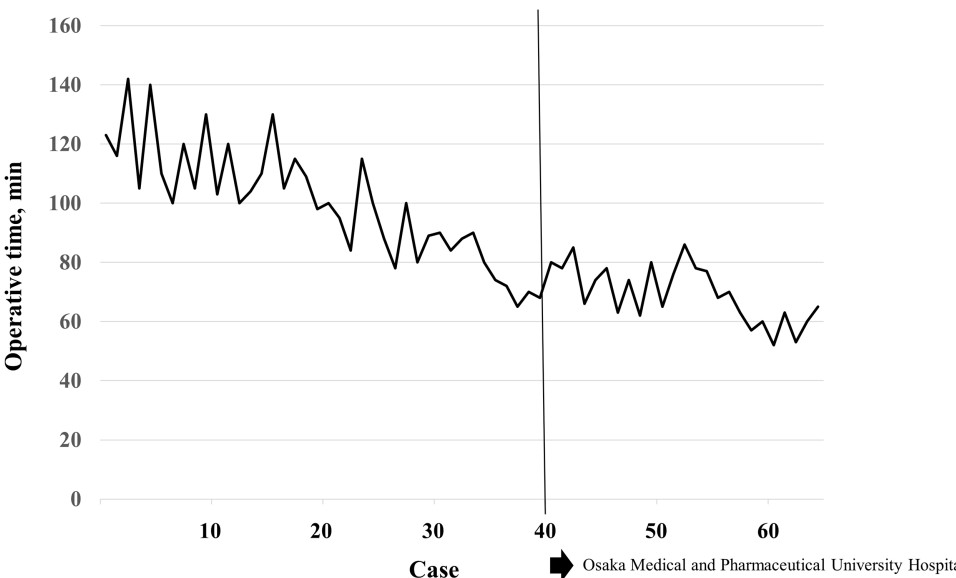

**Fig 5. Learning curve of STEP as performed by the early career surgeon.** The operating time decreased gradually after 20 cases and stabilized after 40 cases. STEP: single-port totally extraperitoneal.

Various RCTs have compared STEP and CTEP [5–10], but they all vary in terms of incision length and site. Five studies reported postoperative pain at 24 hours, but only one study found a significant difference, while the others did not [6–10]. However, some studies have reported significant differences at two or seven days [6,9]. Because the incision length and site have not been standardized, it is not possible to make a general judgment. However, many studies have reported no significant differences in postoperative pain [11]. In contrast, five studies reported the cosmetic outcomes of the operative technique [5,6,8–10]. Satisfaction scores were significantly higher in three studies [5,8,10]. However, it is worth noting that the three studies that reported higher satisfaction scores used different incision sites. Chia et al. and Tran et al. employed a subumbilical transverse incision without the involvement of the umbilicus [5,8]. Conversely, Cardinali et al. reported the use of an umbilical incision [10]. When an incision is made along the umbilical ring below the umbilicus, the shape of the umbilicus remains unchanged but the scar becomes visible. Conversely, an umbilical incision ultimately turns inward and conceals scars. In the study by Cardinali et al., the incision extended beyond the umbilicus, resulting in a visible protruding scar [10]. Although it is still highly cosmetic, for better cosmetic results, it would be desirable to have an incision only within the umbilicus, not extending beyond the umbilicus so that the scar is barely noticeable (Fig 6).

Therefore, we determined that the incision length should be limited to 1–1.5 cm, extending no further than the umbilicus. When subcutaneous dissection was performed, the anterior rectus sheath was exposed and opened using a 1.5 cm incision. Alexis was installed and used to dilate the wound, creating a single free oval hole approximately 2.0 cm × 2.0 cm in diameter, which was large enough to pass three 5-mm devices. The surgical glove port is highly effective for single-port surgery [13,14]. Because the port does not enter the lumen of the wound retractor, surgery is possible with this minimal incision because only the devices pass through the lumen. In addition, this single port facilitates smooth movement of instruments and easy position changes because the port is not fixed.

There are several reports of STEP using surgical gloves [9,15–21], but none using Alexis of XXS size as a wound retractor. Unlike intraperitoneal space, the extraperitoneal space where the initial deployment occurs presents a constrained operative field, so insertion of Alexis is easier when using the smallest inner ring size in this extraperitoneal

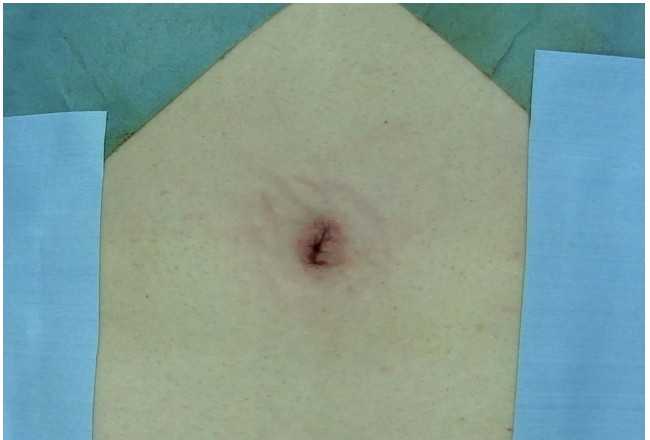

**Fig 6. Postoperative umbilical region.** Although it is still highly cosmetic, for better cosmetic results, it would be desirable to have an incision only within the umbilicus, not extending beyond the umbilicus so that the scar is barely noticeable.

space. Additionally, using the smallest inner ring helps prevent bleeding from the perforating branch of the posterior rectus sheath.

Lee et al. and Kim et al. reported many cases of STEP using the glove method with a total intraumbilical incision [15,19,21]. They reported that STEP using the glove method could be performed safely, even with very small incisions. However, there is a relatively high rate of peritoneal tearing, and there is no mention of mesh insertion methods or benefits. Moreover, the minimization of interference from forceps specific to STEP remains unclear.

In general, the learning curve for TEP is longer and steeper than that for other inguinal hernia repair techniques, primarily because of unfamiliarity with the preperitoneal view and limited working space. Previous studies have reported a learning curve of TEP–40–80 cases [22–25]. Naturally, one might assume that the learning curve for STEP would be even more extended; however, the previously reported range is 40–60, which is not significantly divergent [26,27]. Although a direct institutional comparison with CTEP is not available, these findings suggest that the learning curve of STEP may not be longer than that of CTEP reported in the literature.

In this study, we report the surgical outcomes of an early career surgeon who had previously performed 40 STEP procedures. The patient demonstrated the ability to perform these surgeries safely and without complications requiring treatment, with an average operative time of approximately 60 minutes. Notably, the operative time remained consistent across the 40 cases, showing no significant change compared to previously reported case numbers. The learning curve for the surgeon did not extend, even though he had no prior experience with CTEP and had initiated training with STEP. Therefore, our standardized STEP can be performed without any notable increase in complexity compared with CTEP.

The present study has several limitations. First, this study was retrospective and was conducted by a single surgeon at a single institution, and the potential for case bias may also be considered. Therefore, multicenter studies comparing several surgeons and various patient populations should be conducted to analyze the evolution of the STEP learning curve for STEP. Second, the postoperative pain and cosmetic outcomes were not assessed. However, is this method superior to CTEP in terms of pain and cosmetics compared to CTEP? Thus, an RCT comparing STEP using this method with CTEP is warranted. Third, owing to the absence of long-term follow-up, it remains unknown whether the inguinal hernia recurred. Therefore, long-term follow-up is necessary to determine the rate of hernia recurrence.

In conclusion, STEP using the glove method with a total intraumbilical incision can be safely performed by early career surgeon.

## Supporting information

**S1 File. Raw data – n = 25.**
(XLSX)

**S2 File. Raw data total operative time.**
(XLSX)

## Acknowledgments

The authors thank the staff of the Department of General and Gastroenterological Surgery at the Osaka Medical and Pharmaceutical University. We would also like to thank Editage (www.editage.com) for the English language editing.

## Author contributions

**Conceptualization:** Yoshiro Imai, Mitsuhiro Asakuma.

**Data curation:** Yoshiro Imai.

**Formal analysis:** Yoshiro Imai.

**Investigation:** Yusuke Suzuki.

**Project administration:** Yoshiro Imai.

**Supervision:** Sang-Woong Lee.

**Visualization:** Yoshiro Imai.

**Writing – original draft:** Yoshiro Imai.

**Writing – review & editing:** Yoshiro Imai, Yusuke Suzuki, Mitsuhiro Asakuma, Yoshiharu Miyamoto, Hideki Tomiyama, Sang-Woong Lee.

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
