## [Decision Letter · Decision Letter 0]

12 Aug 2025

Dear Dr. Imai,

Thank you for submitting your manuscript to PLOS ONE. After careful consideration, we feel that it has merit but does not fully meet PLOS ONE’s publication criteria as it currently stands. Therefore, we invite you to submit a revised version of the manuscript that addresses the points raised during the review process.

We look forward to receiving your revised manuscript.

Kind regards,

Iwaho Kikuchi, Ph. D., M.D.

Academic Editor

PLOS ONE

Journal Requirements:

Reviewers' comments:

Reviewer's Responses to Questions

**Comments to the Author**

1. Is the manuscript technically sound, and do the data support the conclusions?

Reviewer #1: Yes

Reviewer #2: Partly

2. Has the statistical analysis been performed appropriately and rigorously?

Reviewer #1: Yes

Reviewer #2: N/A

3. Have the authors made all data underlying the findings in their manuscript fully available?

Reviewer #1: Yes

Reviewer #2: Yes

4. Is the manuscript presented in an intelligible fashion and written in standard English?

Reviewer #1: Yes

Reviewer #2: Yes

Reviewer #1: 1. Were pain scores, return-to-work time, or patient satisfaction evaluated??? These metrics would strongly support claims of cosmetic and postoperative pain benefits.

2. Clarify abbreviations at first mention (e.g., STEP, TEP, CTEP).

3. Discuss the learning curve and how this method fits into surgical training more explicitly.

4.Ethical approval is appropriately stated. However, a brief clarification on how patient anonymity and data protection were maintained would be appreciated.

Reviewer #2: Dear authors,

Thank you for the opportunity to review the paper titled “Standardized Surgical Technique for Single-Port Totally Extraperitoneal Inguinal Hernia Repair Using the Glove Method with an Intraumbilical Incision.”

Below are my comments

Abstract: line 40-42: we present the surgical outcomes at our institution for a surgeon in their eighth year of practice with 40 cases of experience with STEP in this surgical procedure.

Mentioning the volume of cases done before is confusing to the readers. Was the purpose of this to verify that surgeon was beyond the learning curve? If so, please clarify it in the methodology, and perhaps reflect on how that is relevant in general when learning new techniques in the discussion.

Line 99: We present a standardized STEP technique.. – Was this described before? Is it standardized internationally or locally in your service

This statement doesn’t state the aim of this paper clearly. To me this seems to be a paper assessing feasibility of a technique modification with a young surgeon, this involves ease of learning, and safety (patient related outcomes).

Choice of aim will reflect on outcomes. From surgeon’s perspective: discomfort/ease of operating, operative time, #cases during learning curve (reported later), need for conversion or redo

From a patient’s perspective, post-op pain, infection, need for admission, need for re-operation.

It would be helpful to provide a control group (if real-time is not available, then historical) for CTEP in your institution and how it compares to STEP.

Line 117: “From April to October 2023, he performed 29 STEP procedures with the same assistance” - Verify the total number of cases 25 vs 29

Choice of words: it might be better to use standard terminology such as early career surgeon instead of young surgeon. The latter reflects on age more than experience.

**Do you want your identity to be public for this peer review?** For information about this choice, including consent withdrawal, please see our Privacy Policy

Reviewer #1: **Yes: ** Sakarie Mustafe Hidig M.D.

Reviewer #2: **Yes: ** Noora Alshahwani

---

## [Author Response · Author response to Decision Letter 1]

10 Sep 2025

Dear Editor:

We thank you and the reviewers for your thoughtful suggestions and insights. The manuscript has benefited from these insightful suggestions. I look forward to working with you and the reviewers to move this manuscript closer to publication in PLOSONE.

The manuscript has been rechecked and the necessary changes have been made (using track changes mode/red font) in accordance with the reviewers’ suggestions. The responses to all comments have been prepared and attached herewith/given below. We have also had native English speakers check our manuscript.

Thank you for your consideration. I look forward to hearing from you.

Responses to Reviewer Comments

To Reviewer #1

Reviewer’s comment 1

Were pain scores, return-to-work time, or patient satisfaction evaluated??? These metrics would strongly support claims of cosmetic and postoperative pain benefits.

Response to comment 1

Thank you very much for your insightful comment.

This study describes a surgical technique with minimal incisions; however, postoperative pain scores, return-to-work time, and patient satisfaction were not directly evaluated. We agree that these parameters would strongly support the assessment of cosmetic and pain-related benefits. This limitation has been explicitly acknowledged in the revised limitations section.

(lines 300-303)

Reviewer’s comment 2

Clarify abbreviations at first mention (e.g., STEP, TEP, CTEP).

Response to comment 2

Thank you for pointing this out. We have clarified all abbreviations at their first mention (e.g., STEP, TEP, CTEP) in the revised manuscript.

(lines65,66 97,101,102)

Reviewer’s comment 3

Discuss the learning curve and how this method fits into surgical training more explicitly.

Response to comment 3

Thank you for your valuable suggestion. We have expanded the discussion of the learning curve and surgical training. Specifically, we added the following sentence at the end of the Results section:“Figure 6 shows the learning curve of STEP as performed by the early career surgeon. The operative time decreased gradually after 20 cases and stabilized after 40 cases.”

In addition, Figure 6 has been included to illustrate this point more explicitly.

(lines 230-232, 439-441)

Reviewer’s comment 4

Ethical approval is appropriately stated. However, a brief clarification on how patient anonymity and data protection were maintained would be appreciated.

Response to comment 4

Thank you for your valuable suggestion. Patient anonymity and data protection were strictly maintained in accordance with institutional ethics committee approval. To clarify this point, we have added the following sentence to the Materials and Methods section:

“All datasets were anonymized before analysis, and no personal identifiers were used.”

(line 135)

To Reviewer #2

Reviewer’s comment 1

Abstract: line 40-42: we present the surgical outcomes at our institution for a surgeon

in their eighth year of practice with 40 cases of experience with STEP in this surgical

procedure.

Mentioning the volume of cases done before is confusing to the readers. Was the

purpose of this to verify that surgeon was beyond the learning curve? If so, please

clarify it in the methodology, and perhaps reflect on how that is relevant in general

when learning new techniques in the discussion.

Choice of words: it might be better to use standard terminology such as early career surgeon instead of young surgeon. The latter reflects on age more than experience.

Response to comment 1

Thank you for your pertinent suggestion. As you pointed out, mentioning the number of preoperative cases could be confusing, so we have removed it. The intention was to indicate that the surgeon was not beyond the learning curve of conventional TEP, and we have clarified this point in the revised methodology. Furthermore, as you suggested, we have changed the wording from “young surgeon” to “an early career surgeon.” This revision emphasizes that the technique can be mastered even by an early career surgeon.

(lines 78) (lines 85,126,219,287,307)

Reviewer’s comment 2

Line 99: We present a standardized STEP technique. – Was this described before? Is it standardized internationally or locally in your service.

Response to comment 2

Thank you for your valuable suggestion. The term “standardized” in this context refers to the procedure as it has been standardized within our institution, rather than an internationally established standard. To clarify this, we have revised the sentence to:

“We present our institution’s standardized STEP technique.”

Furthermore, we have clarified in the Introduction that the purpose of this paper is to assess the feasibility and safety of this technique when performed by an early career surgeon, with particular emphasis on ease of learning and patient-related outcomes.

(lines 110, 112-114)

Reviewer’s comment 3

Choice of aim will reflect on outcomes. From surgeon’s perspective: discomfort/ease of operating, operative time, #cases during learning curve (reported later), need for conversion or redo From a patient’s perspective, post-op pain, infection, need for admission, need for re-operation. It would be helpful to provide a control group (if real-time is not available, then historical) for CTEP in your institution and how it compares to STEP.

Response to comment 3

Thank you for your thoughtful suggestion. While comparison with CTEP would indeed be valuable, our institution has not implemented CTEP, and therefore a direct comparison—either real-time or historical—is not feasible. Instead, we have emphasized in the revised manuscript that the purpose of this study is to demonstrate the feasibility and safety of STEP as performed by an early career surgeon. In addition, we highlight in the Discussion that, although we cannot provide direct institutional CTEP data, our results suggest that the learning curve of STEP does not appear longer than what has been reported for conventional CTEP in the literature.

(lines 281-286)

Reviewer’s comment 4

“From April to October 2023, he performed 29 STEP procedures with the same assistance” - Verify the total number of cases 25 vs 29

Response to comment 4

I apologize for the previous lack of clarity.

We have performed STEP in 29 cases in total, consisting of 25 unilateral inguinal hernias and 4 bilateral inguinal hernias. In other words, STEP was also applied to bilateral cases. To clarify this point, we have revised the statement accordingly in the manuscript.

(lines 221-222)

---

## [Editor Report · Decision Letter 1]

15 Sep 2025

Standardized Surgical Technique for Single-Port Totally Extraperitoneal Inguinal Hernia Repair Using the Glove Method with an Intraumbilical Incision

PLOS ONE

Dear Dr. Imai,

Thank you for submitting your manuscript to PLOS ONE. After careful consideration, we feel that it has merit but does not fully meet PLOS ONE’s publication criteria as it currently stands. Therefore, we invite you to submit a revised version of the manuscript that addresses the points raised during the review process.

We look forward to receiving your revised manuscript.

Kind regards,

Iwaho Kikuchi, Ph. D., M.D.

Academic Editor

PLOS ONE
---

## [Author Response · Author response to Decision Letter 2]

15 Sep 2025

Thank you very much for your comment.

As no specific references were recommended by the reviewer, we did not make any additional citations.

---

## [Editor Report · Decision Letter 2]

23 Sep 2025

Standardized Surgical Technique for Single-Port Totally Extraperitoneal Inguinal Hernia Repair Using the Glove Method with an Intraumbilical Incision

PONE-D-25-16954R2

Dear Dr. Imai,

We’re pleased to inform you that your manuscript has been judged scientifically suitable for publication and will be formally accepted for publication once it meets all outstanding technical requirements.

Kind regards,

Iwaho Kikuchi, Ph. D., M.D.

Academic Editor

PLOS ONE

Additional Editor Comments (optional):

Thank you for your thorough revisions and thoughtful responses to the reviewers’ and editorial comments. The manuscript has been carefully reviewed and is now considered acceptable for publication in PLOS ONE. We appreciate your contribution and look forward to sharing your work with the broader scientific community.
---

## [Editor Report · Acceptance letter]

PONE-D-25-16954R2

PLOS ONE

Dear Dr. Imai,

I'm pleased to inform you that your manuscript has been deemed suitable for publication in PLOS ONE. Congratulations! Your manuscript is now being handed over to our production team.

Kind regards,

on behalf of

Dr. Iwaho Kikuchi

Academic Editor

PLOS ONE